# Stacked Semantics-Guided Attention Model for Fine-Grained Zero-Shot Learning

**Yunlong Yu, Zhong Ji**[*]
School of Electrical and Information Engineering
Tianjin University
{yuyunlong,jizhong}@tju.edu.cn

**Yanwei Fu**
School of Data Science
Fudan University
AITRICS
yanweifu@fudan.edu.cn

**Jichang Guo, Yanwei Pang**
School of Electrical and Information Engineering
Tianjin University
{jcguo,pyw}@tju.edu.cn

**Zhongfei (Mark) Zhang**
Computer Science Department
Binghamton University
zhongfei@cs.binghamton.edu

## Abstract

Zero-Shot Learning (ZSL) is generally achieved via aligning the semantic relationships between the visual features and the corresponding class semantic descriptions. However, using the global features to represent fine-grained images may lead to sub-optimal results since they neglect the discriminative differences of local regions. Besides, different regions contain distinct discriminative information. The important regions should contribute more to the prediction. To this end, we propose a novel stacked semantics-guided attention ($S^2GA$) model to obtain semantic relevant features by using individual class semantic features to progressively guide the visual features to generate an attention map for weighting the importance of different local regions. Feeding both the integrated visual features and the class semantic features into a multi-class classification architecture, the proposed framework can be trained end-to-end. Extensive experimental results on CUB and NABird datasets show that the proposed approach has a consistent improvement on both fine-grained zero-shot classification and retrieval tasks.

## 1 Introduction

Traditional object classification tasks require the test classes to be identical or a subset of the training classes. However, the categories in the reality have a long-tailed distribution, which means that no classification model could cover all the categories in the real world. Targeting on extending conventional classification models to unseen classes, Zero-Shot Learning (ZSL) [7, 8, 18, 29, 30] has attracted a lot of interests in the machine learning and computer vision communities.

The current approaches formulate ZSL as a visual-semantic alignment problem. In these approaches, an image is represented with its global features. Despite good performances on coarse-grained datasets (e.g., Animal with Attribute dataset [10]), the global features have limitations on fine-grained datasets since more local discriminative information is required to distinguish classes. As illustrated in Fig. 1, the global features only capture some holistic information, on the contrary, the region features capture more local information that is relevant to the class semantic descriptions. Consequently, the global image representations may fail in fine-grained ZSL.

---

[*]The corresponding author.

When trying to recognize an image from unseen categories, humans tend to focus on the informative regions based on the key class semantic descriptions. Besides, humans achieve the semantic alignment by ruling out the irrelevant visual regions, and locating the most relevant ones in a gradual way. Motivated by the above observations and the attention mechanisms that can highlight important local information and neglect irrelevant information of an image, we propose a novel stacked attention-based network to integrate both the global and discriminative local features to represent an image via progressively allocating different weights to different local visual regions based on their relevances to the class semantic descriptions.

As shown in Fig. 2, the proposed approach contains an image featurization part, an attention extraction part, and a visual-semantic matching part. For the image featurization part, we extract the local features that retain the crucial spatial information of an image for the subsequent attention part. It should be noted that the region features can also be compressed into a global one by concatenating or averaging all the local region features. The attention extraction part is the core of the proposed framework, which progressively allocates the importance weights to different visual regions based on their relevance to the class semantic features. The visual-semantic matching part is a two-layer neural network to embed both the class semantic features and the integrated visual features of both the global and local weighted visual features into a multi-class classification framework.

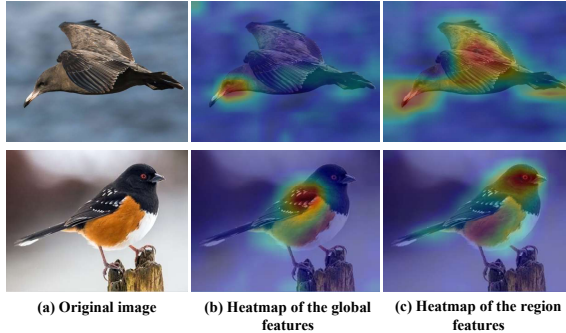

|  (a) Original image | (b) Heatmap of the global features | (c) Heatmap of the region features |

Figure 1: An example of the activation mappings of both the global and region features.

In summary, the contributions of this work are three-fold:

1. We apply the attention mechanism for ZSL to address the issue of irrelevant or noisy information brought by the global features using the local region features of an image. To the best of our knowledge, this is the first work to apply attention mechanism for ZSL.

2. To effectively obtain the attention map to distribute weights for local region features, we propose a stacked attention network guided by the class semantic features for ZSL. It integrates both the global visual features and the weighted local discriminant features to represent an image by allocating different weights for different local regions based on their relevances to the class semantic features.

3. The whole proposed framework can be trained end-to-end by feeding both the visual representations and the class semantic features into a multi-class classification architecture.

In the experiments, we evaluate the proposed $S^2$GA framework for fine-grained ZSL on two bird datasets: Caltech UCSD Birds-2011 (CUB) [22], and North America Birds (NABird) [21]. The experimental results show that our approach significantly outperforms the state-of-the-art methods with large margins on both zero-shot classification and retrieval tasks.

## 2   Related Work

### 2.1   Region features-based Zero-Shot approaches

A limitation of most existing ZSL approaches for fine-grained datasets is to use global features to represent the visual images. This may feed irrelevant or noisy information to the prediction stage. Recently, [1, 6, 33] adopt local region features to represent images that are more relevant to the class semantic descriptions. Specifically, [6] proposed to learn region-specific classifier to connect text terms to its relevant regions and suppress connections to non-visual text terms without any part-text annotations. By concatenating the feature vectors of each visual region as the visual representations of an image, Zhu *et al*. [33] proposed a simple generative method to generate synthesised visual features using the text descriptions about an unseen class. Inspired by the effectiveness of region

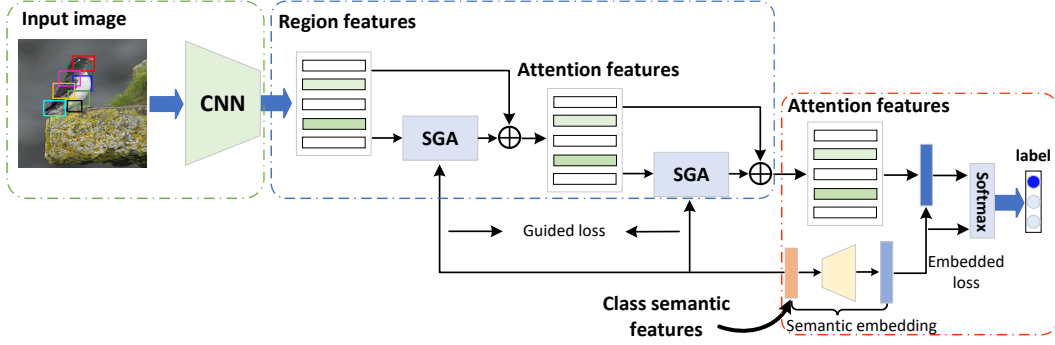

Figure 2: The framework of the proposed $S^2GA$ approach. The green box denotes the image featurization part; the blue box is two-layer Semantics-Guided Attention (SGA) part to distribute different weights to different relevant visual features based on the class semantic descriptions; the red box denotes the visual-semantic matching part to jointly embed both the class semantic features and the integrated features of both the global and local weighted features into a multi-class classification framework.

features for ZSL, we also use the region features in this work. Different from [1, 6, 33], we argue that important regions should contribute more to the prediction and design an attention method to distribute different weights for different regions according to their relevance with class semantic features, and integrate both the global visual features and the weighted region features into more semantics-relevant features to represent images.

## 2.2 Attention Model

The aim of the attention mechanisms is to either highlight important local information or alleviate the issue of irrelevant or noisy information brought by the global features. Due to their validity and generality, the attention mechanisms have been widely adopted in various computer vision tasks, e.g., object classification [4], machine translation [9], image caption [13, 27], and visual question answering [24, 28].

To the best of our knowledge, there is no work to apply the attention mechanism to ZSL task. In this work, we design a stacked attention network to assign different importance weights to features of different local regions to obtain a more semantics-relevant feature representation. One of the most closely related work, Yang *et al.* [26] proposed a stacked attention network for visual question answering, which directly uses the question to search for the related regions to the answer. However, for ZSL task, since no corresponding class semantic descriptions are provided for testing images, the current attention mechanism could not be applied to it directly. To this end, we propose to indirectly learn the attention maps to weight different regions guided by the class semantic descriptions during training. This is the novelty of the proposed attention method. Besides, we also integrate both the weighted region features and the global features to represent image features since the corresponding class semantic descriptions contain both global and local information. In summary, our stacking based attention learning enables learning a hierarchical representation of the attention from both the global and local features, which was ignored by the existing studies in the attention learning literature and is also significantly different from the method at [23] that does not use stacking mechanism.

## 3 Semantics-guided Attention Networks

The visual-semantic matching that measures similarities between the visual and class semantic features is a key to address ZSL. However, the visual features extracted from the original images and the class semantic features are located in different structural spaces, a simple matching method may not align the semantics well. To narrow the semantic gap between the visual and the class semantic modalities, we propose a semantic-guided attention approach to use the class semantic descriptions to guide the local region features to obtain more semantic-relevant visual features for the subsequent visual-semantic matching. The semantic-guided attention mechanism pinpoints the

regions that are highly relevant to the class semantic descriptions and filters out the noisy regions. The overall architecture of the proposed semantics-guided attention networks is shown in Fig. 2. In this section, we first describe the proposed S$^2$GA model and the visual-semantic matching model, and then apply it to fine-grained ZSL.

## 3.1 Stacked semantics-guided attention networks

Given the local features of an image and its corresponding class semantic vector, the attention networks distribute different weights for each visual region vector via multi-step attention layers, and integrate both the global and weighted local features to obtain more semantics-relevant representations for images. Specifically, we propose an attention approach using the local region features to gradually filter out noises and weight the regions that are highly relevant to the class semantic descriptions via multiple (stacked) attention layers. It consists of multi-step attention layers to generate the distribution weights for relevant regions. Each attention map is obtained to measure the relevance between the region features and the class semantic features under the guidance of the class semantic features, so we call our approach stacked semantics-guided attention(S$^2$GA) model.

Fig. 3 shows the illustration of a semantic guided-attention layer. Given the image local feature representations $\mathbf{V}_I$ and its corresponding class semantic vector $\mathbf{s}$, the attention map is obtained with two separated networks. The first network is denoted as local embedding network, which feeds the local feature representations into a latent space through a simple two-layer neural network. The second network is named as semantic-guided network. It first compresses all the visual region features to an integrate vector $\mathbf{v}_G$ and then feeds it into the same latent space with a three-layer neural network. Specifically, the output of the mid-layer is

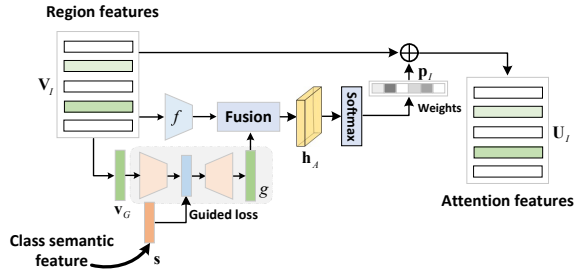

Figure 3: An illustration of a semantics-guided attention (SGA) layer.

forced to be close to the corresponding class semantic feature. In this way, the class semantic information is embedded into the network, which guides to obtain the attention map. Then, a softmax function is used to generate the attention distribution over the regions of the image:

$$\mathbf{h}_A = \tanh(f(\mathbf{V}_I) \oplus g(\mathbf{v}_G)), \tag{1}$$

$$\mathbf{p}_I = \text{softmax}(\mathbf{W}_P \mathbf{h}_A + b_P), \tag{2}$$

where $\mathbf{V}_I \in \mathbb{R}^{p \times m}$, $p$ is the feature dimensionality of each region and $m$ is the number of the image regions, $\mathbf{v}_G \in \mathbb{R}^p$ is the fused visual image vector that is the uniform average of all the image region vectors $\mathbf{V}_I$, $\oplus$ denotes the multiplication between each column of the matrix and the vector, which is performed by element-wise multiplying each column of the matrix by the vector. $\mathbf{h}_A \in \mathbb{R}^{d \times m}$ is the fused vector in the latent space, $\mathbf{p}_I \in \mathbb{R}^m$ is an $m$ dimensional vector, which corresponds to the attention probability of each image region, $f$ and $g$ are two different networks:

$$f(\mathbf{v}_I) = h(\mathbf{W}_{I,A} \mathbf{v}_I), \tag{3}$$

$$g(\mathbf{v}_G) = h(\mathbf{W}_{G,A} h(\mathbf{W}_{G,S} \mathbf{v}_G)), \tag{4}$$

where $h$ is a nonlinear function (we use Rectified Linear Unit (ReLU) in the experiments). $\mathbf{W}_{I,A} \in \mathbb{R}^{d \times p}$, $\mathbf{W}_{G,S} \in \mathbb{R}^{q \times p}$, $\mathbf{W}_{G,A} \in \mathbb{R}^{d \times q}$ are the learned parameters, $q$ and $d$ are the dimensionalities of the class semantic space and the latent space, respectively. To embed the class semantic information into the attention network, the output of the second-layer of $g$ is forced to be close to the corresponding class semantic features, which is formulated as:

$$\min Loss_G = \|h(\mathbf{W}_{G,S} \mathbf{v}_G) - \mathbf{s}\|_2. \tag{5}$$

Based on the attention distribution, we obtain the weighted feature vector of each image region, $\tilde{\mathbf{v}}_i = p_i \mathbf{v}_i$, where $\mathbf{v}_i$ is the feature vector of $i$-th region. We then combine the weighted vector $\tilde{\mathbf{v}}_i$ with the previous region feature vector to form a refined region vector $\mathbf{u}_i = \tilde{\mathbf{v}}_i + \mathbf{v}_i$. $\mathbf{u}_i$ is regarded as a refined vector since it encodes both visual information and the class semantic information.

Compared with the approaches that simply use the global image vector, the attention method constructs a more informative $\mathbf{u}$ since higher weights are put on the visual regions that are more relevant to the class semantic descriptions. However, for some complicated cases, a single attention network is not sufficient to locate the correct region for class semantic descriptions. Therefore, we iterate the above semantics-guided attention process using multiple attention layers, each extracting more fine-grained visual attention information for class semantic descriptions. Specifically, for the $k$-th attention layer, we compute:

$$\mathbf{h}_A^k = \tanh(f(\mathbf{U}_I^k) \oplus g(\mathbf{u}_G^k)), \tag{6}$$

$$\mathbf{p}_I^k = \mathrm{softmax}(\mathbf{W}_P^k \mathbf{h}_A^k + b_P^k), \tag{7}$$

where $\mathbf{U}_I^0$ and $\mathbf{u}_G^0$ are initialized to be $\mathbf{V}_I$ and $\mathbf{v}_G$, respectively. Then the weighted image region vector is added to the previous image region feature to form a new image vector:

$$\tilde{\mathbf{u}}_i^k = p_i^k \mathbf{u}_i^k, \tag{8}$$

$$\mathbf{u}_i^k = \tilde{\mathbf{u}}_i^k + \mathbf{u}_i^k. \tag{9}$$

It should be noted that in every attention layer the class semantic information is embedded into the network. We repeat this process $K$ times to obtain $\mathbf{u}_G^K$. By integrating all the region features, we obtain the final image representations $\mathbf{u}_G$ for the embedding network.

## 3.2 Visual-semantic matching model

To connect the visual features and class semantic features, we use a two-layer network to embed the class semantic feature into the visual space:

$$\mathbf{v}_s = h(\mathbf{W}_E \mathbf{s} + \mathbf{b}_E), \tag{10}$$

where $\mathbf{W}_E \in \mathbb{R}^{p \times q}$, $\mathbf{b}_E \in \mathbb{R}^q$ are the embedding matrix and bias, and $h$ denotes the ReLU function. To align the common semantic information between the visual space and the class semantic space, the differences between the attention visual feature $\mathbf{u}_G$ and its corresponding feature embedding $\mathbf{v}_s$ of class semantic descriptions should be small, and the objective function is formulated as a squared loss:

$$\min Loss_A = \|\mathbf{v}_s - \mathbf{u}_G\|_2. \tag{11}$$

It should be noted that the contrastive loss is also an alternative to align the visual-semantic interactions, which is specifically formulated as:

$$\min Loss_A = max\{0,\ m + d(\mathbf{v}_s; \mathbf{u}_G) - d(\mathbf{v}_s^{neg}; \mathbf{u}_G)\}, \tag{12}$$

where $m$ is a margin which we fix to 0.1 in the experiments; $\mathbf{v}_s^{neg}$ is the synthesized visual features from the other class semantic features; $d(\mathbf{i}; \mathbf{j})$ is a metric function that measures the distance between vectors $\mathbf{i}$ and $\mathbf{j}$. While in our experiments we use a square Euclidean distance $d(\mathbf{i}; \mathbf{j}) = \|\mathbf{i} - \mathbf{j}\|_2$, any other differentiable distance functions can also be used here.

Considering that the approach with the squared loss is more efficient, if not specified, $\mathrm{S}^2\mathrm{GA}$ is referred to the approach with the squared loss.

A multi-class classifier with softmax activation, which is trained on the final visual attention features as well as the class semantic features, predicts the class label of the input image. The predicted label is the index with the maximum probability:

$$c^* = arg \max \mathbf{p}_c,\ \ s.t.\ \ \mathbf{p}_c = \mathrm{softmax}(\mathbf{V}_S \mathbf{u}_G), \tag{13}$$

where $\mathbf{V}_S$ is the embedding matrix of the collection of all the seen class semantic features, which is obtained in Eq. 10.

## 3.3 Apply to ZSL

Given a set of semantic features $\mathbf{S}_T$ of candidate classes and a test instance $\mathbf{v}_t$, ZSL is achieved via three steps. First, the test instance is fed into the attention network to obtain the attention feature $\mathbf{u}_t$. Then, the semantic features are embedded into the visual space in Eq. 10 to obtain $\mathbf{V}_T$. After that, the classification of the test instance is achieved by simply calculating its distance to the semantic embedding features $\mathbf{V}_T$ in the visual space:

$$c_t^* = \arg \min_t \mathcal{D}(\mathbf{u}_t, \mathbf{V}_T) \tag{14}$$

| Method | F | SI | Performance |
|---|---|---|---|
| MFMR-joint[†] [25] | $\mathcal{V}$ | $\mathcal{A}$ | 53.6 |
| Deep-SCoRe [12] | $\mathcal{V}$ | $\mathcal{A}$ | 59.5 |
| SynC$^{struct}$ [5] | $\mathcal{G}$ | $\mathcal{A}$ | 54.4 |
| DEM [32] | $\mathcal{G}$ | $\mathcal{A}$ | 58.3 |
| ESZSL [15] | $\mathcal{G}$ | $\mathcal{A}$ | 53.1 |
| Relation-Net [19] | $\mathcal{G}$ | $\mathcal{A}$ | 62.0 |
| SJE [3] | $\mathcal{G}$ | $\mathcal{A}/\mathcal{W}$ | 55.3/28.4 |
| MCZSL [1] | GTA | $\mathcal{A}/\mathcal{W}$ | 56.5/32.1 |
| S$^2$GA | GTA | $\mathcal{A}/\mathcal{W}$ | 75.3/46.9 |
| S$^2$GA-*CL* | GTA | $\mathcal{A}/\mathcal{W}$ | **76.8/48.2** |

Table 1: Performance evaluation on CUB in classification accuracy (%). S$^2$GA-*CL* is the approach with the contractive loss; $\mathcal{V}$ and $\mathcal{G}$ are short for VGGNet and GoogleNet feature representations. $\mathcal{A}$ and $\mathcal{W}$ are short for attribute space and Word2Vec space, respectively. MFMR-joint[†] is a transductive approach in which the testing instances are available in the training stage.

## 4 Experiments

In this section, we carry out several experiments to evaluate the proposed S$^2$GA networks on both zero-shot classification and zero-shot retrieval tasks.

### 4.1 Experimental setup

**Datasets:** Following [6], we conduct experiments on two fine-grained bird datasets, CUB [22] and NABirds [21]. Specifically, CUB dataset contains 200 categories of bird species with a total of 11,788 images. Each category is annotated with 312 attributes. Besides, the local regions of the bird in each image are annotated with locations by experts. Thus, we can either directly extract the local image features using the ground-truth location annotations or indirectly extract the local image features using the SPDA-CNN framework [31] to detect the important regions followed a sub-network that uses a 3×3 ROI to pool each region for a 512-d feature. We call the features based on these two different strategies as "GTA" and "DET", respectively. Specifically, we extract the features of 7 local regions to represent each CUB image. The dimensionality of each region is 512. These 7 regions are "head", "back", "belly", "breast", "leg", "wing", and "tail". Considering that the class semantic descriptions are easily available for CUB dataset, we conduct experiments using class-level attribute, Word2Vec [11] and Term Frequency-Inverse Document Frequency (TF-IDF) [16] feature vector as class semantic features, respectively. The dimensionalities of the attribute, Word2Vec and TF-IDF are 312, 400 and 11,083, respectively. Compared with CUB dataset, NABirds dataset is a larger dataset. It consists of 1,011 classes with 48,562 images. As the same as that in [6], we obtain the final 404 classes after merging the leaf node classes into their parents. The semantic descriptions of each category is an article collected from Wikipedia, and Term Frequency-Inverse Document Frequency (TF-IDF) [16] feature vector is then extracted to represent the class semantic features. The dimensionality of TF-IDF for NABird dataset is 13,585. Since no "leg" annotations of NABird dataset are available, we extract the features of the remaining six visual regions to represent the local visual features. In order to improve the training efficiency, we use Principal Component Analysis (PCA) to reduce the TF-IDF dimensionality of CUB and NABird datasets to 200 and 400, respectively. All the class semantic features are scaled into [0, 1] with the standard normalization.

**Implementation Details:** In our system, the dimensionality $d$ of the hidden layer and the batch size are set to 128 and 512, respectively. We directly optimize the sum of these three objective functions without weights as we found empirically adding weights did not improve the performances. The whole architecture is implemented on the Tensorflow and trained end-to-end with fixed local visual features [2]. For optimization, the RMSProp method is used with a base learning rate of $10^{-4}$. The architecture is trained for up to 3,000 iterations until the validation error has not improved in the last 30 iterations.

| Method | CUB | | NABird | |
|---|---|---|---|---|
| | SCS | SCE | SCS | SCE |
| ESZSL [15] | 28.5 | 7.4 | 24.3 | 6.3 |
| ZSLNS [14] | 29.1 | 7.3 | 24.5 | 6.8 |
| SynC$^{fast}$ [5] | 28.0 | 8.6 | 18.4 | 3.8 |
| ZSLPP [6] | 37.2 | 9.7 | 30.3 | 8.1 |
| GAA [33] | **43.7** | 10.3 | 35.6 | 8.6 |
| S$^2$GA-DET (Ours) | 42.9 | **10.9** | **39.4** | **9.7** |

Table 2: The per-class average Top-1 accuracy (in %) on CUB and NABird datasets with two different split settings using DET as visual representations and TF-IDF as class semantic features. We directly copy the performance results of the competitors from [6] and [33].

## 4.2 Traditional ZSL

### 4.2.1 Comparison with state-of-the-art approaches

Compared with NABird dataset, CUB dataset is a well-known dataset for ZSL. Thus, we first conduct experiments on CUB dataset for comparing with the state-of-the-art approaches. For an easy comparison available with the existing approaches, we use the same split as in [2] with 150 classes for training and 50 disjoint classes for testing. Eight recently published approaches are selected for comparison. Specifically, MCZSL [1] directly uses region annotations to extract image features, and the rest approaches employ two popular CNN architectures to extract image features, i.e., VGGNet [17] and GoogleNet [20]. As for the class semantic representations, we use the attributes provided by the CUB dataset and Word2Vec [11] of each class name extracted with the unsupervised language processing technology. We report the per-class average Top-1 accuracies of different approaches in Table 1.

Although performance results in Table 1 vary drastically with different visual representations, it is clear that the proposed approach outperforms all the existing methods both with attribute annotations and Word2Vec. Specifically, the proposed S$^2$GA approach using GTA achieves impressive gains against the state-of-the-art approaches for both semantic representations: 13.3% on the attribute and 14.8% on Word2Vec, respectively. Besides, we observe that S$^2$GA-*CL* performs slightly better than S$^2$GA, which indicates that the contrastive loss may capture more discriminative information.

To compare with the existing methods more fairly, we select five state-of-the-art approaches that use both the same visual representations and class semantic representations on both CUB and NABird datasets. All the competitors use the same settings and the features as ours. Specifically, we use the VGGNet features based on the detected regions to represent visual image and TF-IDF to represent class semantic features. All the region vectors are concatenated in order into a long vector to represent each image. Those features are provided in [6], which are obtained in [3]. Following [6, 33], we evaluate the approaches on two split settings: Super-Category-Shared (SCS) and Super-Category-Exclusive (SCE). These two different splits differ in how close the seen classes are related to the unseen ones. Specifically, in the SCS-split setting, there exists one or more seen classes belonging to the same parent category of each unseen class. This is the traditional ZSL split setting on CUB dataset. On the contrary, in the SCE-split setting, the parent categories of unseen classes are exclusive to those of the seen classes. Compared with that of SCS-split, the relevance between seen and unseen classes of SCE-split is minimized, which brings more challenges for knowledge transfer.

Table 2 shows the classification performances of each split setting on both CUB and NABird datasets. From the results, we observe that the proposed S$^2$GA approach beats all the competitors on both two datasets under two different split settings except a slight 0.8% worse than GAA [33] under SCS split setting on CUB dataset. Specifically, it obtains 3.8% and 1.1% improvements against the current best approach [33] on NABird dataset under both SCS and SCE split settings. Besides, we observe that the classification accuracies of SCE are dramatically smaller than those of SCS, which indicates that the relativity between the seen classes and unseen classes impacts the classification performances a lot. This is a reasonable phenomenon since the knowledge is easily transferred if some related classes or parent classes of unseen classes are available for training.

| Method | CUB | | | NABird |
|---|---|---|---|---|
| | $\mathcal{A}$ | $\mathcal{W}$ | $\mathcal{T}$ | $\mathcal{T}$ |
| baseline | 62.5 | 38.4 | 38.6 | 31.6 |
| one-attention layer | 67.1 | 40.3 | 41.8 | 36.2 |
| two-attention layer | **68.9** | **41.8** | **42.9** | 39.4 |
| three-attention layer | 68.5 | 41.6 | 42.7 | **39.6** |

Table 3: ZSL performances (in %) of the proposed approach with different attention layers on CUB and NABird datasets. $\mathcal{A}$ - attribute, $\mathcal{W}$ - Word2Vec, $\mathcal{T}$ - TF-IDF. "baseline" is the method without attention mechanism.

| Method | CUB | | NABird | |
|---|---|---|---|---|
| | 50% | 100% | 50% | 100% |
| ESZSL [15] | 27.3 | 22.7 | 27.8 | 20.9 |
| ZSLNS [14] | 29.5 | 23.9 | 27.3 | 22.1 |
| ZSLPP [6] | 42.0 | 36.3 | 35.7 | 31.3 |
| GAA [33] | **48.3** | 40.3 | 37.8 | 31.0 |
| S$^2$GA (Ours) | 47.1 | **42.6** | **42.2** | **36.6** |

Table 4: Zero-shot retrieval mAP (in %) comparison on CUB and NABird datasets. The results of all the competitors are cited from [33]. All the competitors use the same features.

#### 4.2.2 Gains of the attention mechanism

To evaluate the effectiveness of the proposed attention mechanism, we conduct experiments on both CUB and NABird datasets using local features based on detected regions as visual representations under the SCS split setting. We evaluate the methods without attention mechanism and with different layer attention mechanism. We call the method without attention layer as baseline. The comparison results are shown in Table 3.

From the results, we find that the performances of the methods with attention mechanism are much better than those without attention mechanism on both CUB and NABird datasets, which verifies the effectiveness of the attention mechanism. Besides, the stacked attention mechanism can further improve the performances. However, when the attention layer is larger than two, the performances tend to be steady, possibly because there is no further margin of improvement.

### 4.3 Zero-Shot Retrieval

The task of zero-shot retrieval is to retrieve the relevant images from unseen class set related to the specified class semantic descriptions of unseen classes. Here we use the above well trained method to embed both all the images of unseen classes and the class semantic descriptions into the integrated feature space spanned both the global features and the weighted local features where the semantic similarities of the visual and class semantic representations are obtained. For comparing with the competitors fairly, we use both the same feature representations (visual/textual) as well as the same settings as that in [33] where 50% and 100% of the number of images for each class from the whole dataset are ranked based on their final semantic similarity scores.

Table 4 presents the comparison results of different approaches for mean accuracy precision (mAP) on CUB and NABird datasets. Note that except for the case of 50% of the number of images for each class on CUB dataset, the proposed approach beats all the competitors. We argue that it benefits from both the final powerful feature representations based on the proposed S$^2$GA mechanism as well as the efficient alignment between the visual modality and the class semantic modality.

We also visualize some qualitative results of our approach on two datasets, shown in Fig. 4. We observe that the retrieval performances of different classes vary substantially. For those classes that have good performances, their intra-class variations are subtle. Meanwhile, for those classes that have worse performances, their inter-class variations are small. For example, the top-6 retrieval images of class "Indigo Bunting" are all from their ground truth class since their visual features are similar. However, the query "Black-billed Cuckoo" retrieves some instances from its affinal class "Yellow-billed Cuckoo" since their visual features are too similar to distinguish.

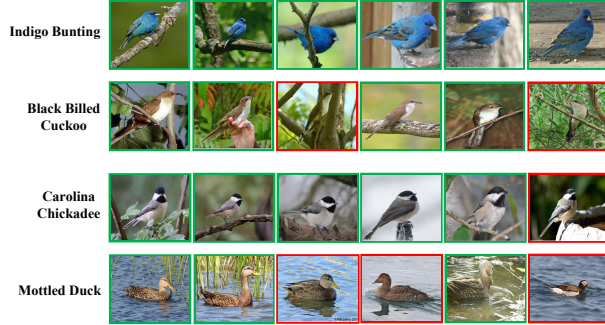

Figure 4: Zero-shot retrieval visualization samples with our approach. The first two rows are classes from CUB dataset, and the rest are classes from NABird dataset. Correct and incorrect retrieved instances are marked in green and red boxes, respectively.

## 5 Conclusion

In this paper, we have proposed a stacked semantics-guided attention approach for fine-grained zero-shot learning. It progressively assigns weights for different region features guided by class semantic descriptions and integrates both the local and global features to obtain semantic-relevant representations for images. The proposed approach is trained end-to-end by feeding the final visual features and class semantic features into a joint multi-class classification network. Experimental results on zero-shot classification and retrieval tasks show the impressive effectiveness of the proposed approach.

**Acknowledgments**

This work was supported by the National Natural Science Foundation of China under Grant 61771329, the National Basic Research Program of China (Grant No. 2014CB340403), the National Natural Science Foundation of China under Grant 61632018. Yunlong Yu also acknowledges the support of China Scholarship Council. The authors are very grateful for NVIDIA's support in providing GPUs that made this work possible.

## Footnotes

[2]https://github.com/ylytju/sga

[3]`https://github.com/EthanZhu90/ZSL_PP_CVPR17`

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
