[Supplementary Material · feedback.pdf]

We thank the AC and the 3 reviewers for the comments and efforts.

# # Reviewer 1

**Q1: The method is reasonable and effective... not a ground-breaking work as adding attention mechanism.**

**A1:** While we appreciatively agree with the fact that attention has been extensively used in the literature, we respectfully emphasize that as motivated in L21-25 and discussed in L86-100, the attention-based feature fusion method is indeed new in the literature and is also supported empirically as reported in the paper.

**Q2: Applying algorithms to AwA datasets.**

**A2:** As stated in the paper, our method is designed to solve fine-grained classification problems that require supervision of part annotations. AwA is a coarse-grained dataset without part annotations and thus is not appropriate.

**Q3: Referring to Wang et al. CVPR-17; and the discussion about the differences.**

**A3:** We have added the reference Wang et al. CVPR-17 into the discussion in the paper. The details are in the **related work** part.

Our stacking-based attention learning enables learning a hierarchical representation of the attention from both the global and local features, that was ignored by the existing studies in the attention learning literature and is also significantly different from the method at [Wang et al. CVPR-17] that does not use stacking mechanism.

[Wang et al. CVPR-17] Peng Wang, Lingqiao Liu, Chunhua Shen, Zi Huang, Anton van den Hengel, and Heng Tao Shen. Multi-attention network for one shot learning. In CVPR, pages 22–25, 2017.

# # Reviewer 2

**Q1: Trade-offs of the scalability to a large number of candidate categories?**

**A1:** We appreciatively agree that such conditioning may restrict the scalability. However, in practice, images typically do not have many regions. Further, in the case of images with many regions, there are approximation methods such as parallelization to significantly speed up the computation without significantly limiting the performance per our studies.

**Q2: How's the initial fused image regions computed? How about the baseline model?**

**A2:** In both the baseline model and our approach, the region vectors are concatenated into a long vector in order. We have added the explanation in the **experiments** part.

**Q3: There appear to be 3 losses ... Do you directly optimize the sum of these 3 objectives?**

**A3:** Yes, we directly optimize the sum of these three objectives without weights as we found empirically adding weights did not improve results. We have explained it in the **experiments** part.

**Q4: In Figure 3 referring as "c" but use "s".**

**A4:** We have fixed this typo in the paper.

**Q5: Experimented with alternatives to L2 for matching the 2 modalities? For example, using a contrastive loss?**

**A5:** We greatly appreciate this comment and conducted the experiments with the contrastive loss, leading to performances of 76.8% and 48.2% on CUB with attributes and wordvec, respectively, slightly better than those with L2 loss. We have added the details about the contrastive loss in the paper.

It should be noted that the contrastive loss is also an alternative to align the visual-semantic interactions, which is specifically formulated as:

$$\min Loss_A = max\{0, m + d(v_s; u_G) - d(v_s^{neg}; u_G)\} \tag{12}$$

where $m$ is a margin which we fix to 0.1 in the experiments; $\mathbf{v}_s^{neg}$ is the synthesized visual features from the other class semantic features; $d(\mathrm{i};\mathrm{j})$ is a metric function that measures the distance between vectors i and j. While in our experiments we use a square Euclidean distance $d(\mathrm{i};\mathrm{j}) = \left\| \mathrm{i} - \mathrm{j} \right\|_2$, any other differentiable distance functions can also be used here.

**Q6: Referring the attention papers.**

**A6**: We have added the reference in the **related work** part.

Melvin Johnson, Mike Schuster, Quoc V Le, Maxim Krikun, Yonghui Wu, Zhifeng Chen, Nikhil Thorat, Fernanda Viégas, Martin Wattenberg, Greg Corrado, et al. Google's multilingual neural machine translation system: Enabling zero-shot translation. ACL, 5(1):339–351, 2017.

# # Reviewer 3

**Q1: ... what is the feature extractor used?... Comparing resnet / descent with vgg / googlenet not fair.**

**A1**: We appreciatively agree and will add more descriptions/explanations. We adopt SPDA-CNN framework with VGG-16 to detect bird-parts. The part regions are detected with the SPDA-CNN framework based VGG-16, and then passed to a sub-network which uses a 3 x3 ROI to pool these regions for a 512-d feature. We did have the experiments using the same features also supporting that our method is better but had to drop them due to space limitations.

**Q2: The presentation of this paper can be further improved.**

**A2:** We have followed the suggestions and reorganize and improve the presentation.

**Q3: Figure 3: class semantic feature should be labeled as "s" instead of "c"?**

**A3**: We have fixed this typo in the paper.

**Q4: equation 1: how v_G is fused from V_I? please specify.**

**A4:** v_G is the fused visual image vector that is the uniform average of all the image region vectors V_I. We have added this explanation in the 3.1 part.

**Q5: - equation 5: s is coming from textual representations (attribute / word to vec / PCA'ed TFIDF). It might have positive/negative values? However the first term h(W_{G,S}, v_G) is post ReLU and can only be non-negative?**

**A5:** We have preprocessed the class semantic vector into [0,1] with the standard normalization.

**Q6: Would having a scaling variable before attention weight help?**

**A6:** Yes, the refined vector scales and having scaling variable before attention weight helps.

**Q7: Equation 11: v_s and u_G are both outputs from trained-network, and they are not normalized? So minimize L-2 loss could be simply reducing the magnitude of both vectors?**

**A7:** No. The v_s and u_G are outputs of the networks whose the final layers are both ReLU, which normalizes the value of the input into the same range, so L2 loss can reduce the magnitude of both vectors.

**Q8: Line 201: the dimensionality of each region is 512: using which feature extractor?**

**A8:** We use the SPDA-CNN framework with VGG-16, and following a sub-network which uses a 3x3 ROI to pool the detected region into 512-d features.

**Q9: Section 4.2.2: comparing number of attention layers is a good experiment. Another baseline could be not using Loss_G? So attention is only guided by global feature vector.**

**A9:** We followed the comment and added the baselines with the results of text features 38.6% and 31.6% on CUB and NABird, respectively, again supporting the method.

**A10: Table 4: what are the visual / textual representations used in each method? otherwise it is unclear whether the end-to-end performance gain is due to proposed attention model.**

**A10:** Yes. In our experiments, the same feature representations (visual/textual) and the same settings are used for the comparative approaches. We have added the details in the **experiments** part.

The code of this paper is released in https://github.com/ylytju/sga