[Reviews · NeurIPS 2018]

Reviewer 1



Summary ======================= This paper presents a stacked semantics-guided attention (S2GA) model for improved zero-shot learning. The main idea of this paper is that important regions should contribute more to the prediction. To this end, the authors design an attention method to distribute different weights for different regions according to their relevance with class semantic features and integrate both the global visual features and the weighted region features into more semantics-relevant features to represent images. Strengths ======================= + The method is well motivated. The presentation of the method is clear. + Using stacked attention for zero-shot learning seems to be a new idea (I do not check exhaustively). Although stacked attention network has been presented in [26]. The authors explain the differences in Sec 2.2. + Consistent improvement is shown on fine-grained image datasets such as CUB and NABird datasets. + The ablation study in Table 3 convincingly shows the effect of the proposed attention mechanism. Weaknesses ======================= 1) The method is reasonable and effective for the targeted task. However, it is not a ground-breaking work as adding attention mechanism to different visual tasks becomes quite common these days. 2) It is not clear if the method would perform equally well on other ZSL datasets (e.g., Animal with Attribute dataset). 3) The benefit of stacking attention is not entirely clear, although positive performance is shown when more attention layers are used. It would be good to discuss what kind of attention has been learned in the two attention stages. Perhaps more heatmap images like Fig 1 can be presented. 4) The following one-shot-learning paper that uses attention is relevant. It also uses attention guided by semantics. In addition, they have multiple attention maps without stacking. A discussion is needed on the differences and why stacking is essential. Peng Wang et al., Multi-attention Network for One Shot Learning, CVPR 2017.

Reviewer 2



This paper presents an approach to zero-shot learning through the use of a stacked attention mechanism. Given region-based features for an image and a class semantic feature, the semantic feature acts as a controller to an attention mechanism that computes a weighted average based on the relevance of each of the region features to the class semantic feature. This module can be repeated multiple times, adding the previous region features to the newly weighted features iteratively. In order to make class predictions, the authors incorporate a visual-semantic matching model that embeds the final region-based features with the transformed class semantic features, followed by a softmax over classes. At test time, a collection of semantic features for candidate classes are fed into the model, producing a distance to the processed region features. The class with the smallest distance is then assigned to the image. The authors perform experiments on standard ZSL benchmarks, achieving strong performance relative to existing methods. Furthermore, the authors validate the utility of their attention mechanism, showing that using 2 layers outperforms a single layer and a baseline model without attention. The paper is well written and the method description is fairly clear. Overall a strong contribution. However, I am not deeply familiar with ZSL experimental protocols so I cannot comment confidently on the intricacies of the evaluations. A few questions / comments: - Since computing region features is conditional on the class semantic features, this would mean that inference would be much slower than an unconditional method whose image representations are not a function of the semantic features (e.g. uniform average). Presumably, this would restrict the scalability of your method in a setting with a large number of candidate categories. Could you discuss these trade offs? - How is the initial fused image regions computed? Is it just a uniform average? Is this what’s done in the baseline model? - There appear to be 3 losses: 2 based on L2 (Loss_G and Loss_A) and a third classification (presumably cross-entropy) loss. Do you directly optimize the sum of these 3 objectives? How do you weight the L2 losses with the classification loss? - In Figure 3 you refer to the class semantic feature as “c” but use “s” throughout the paper. - Have you experimented with alternatives to L2 for matching the 2 modalities? For example, using a contrastive loss? - While attention may not have been applied to ZSL, it has been used in the related areas of one/few shot learning (e.g. Matching networks). Google's zero-shot translation paper (Johnson et al) may also qualify.

Reviewer 3



This paper aims at using attention on image regions to improve fine grained zero shot learning task. The proposed method is build up on existing zero shot learning frame work with a novel stacked attention branch, which matches global image and text representation and use it to weight regional features. The proposed method shows better performance in two bird datasets. Strengths: - The idea to put attention into zero shot learning task, especially first regresses global image feature to textual representation, then use it to guide attention, is new. - Proposed method shows better performance in two datasets by a significant margin, especially in zero-shot retrieval task. Weaknesses: - This paper misses a few details in model design and experiments: A major issue is the "GTA" / "DET" feature representation in Table 1. As stated in section 4.1, image regions are extracted from ground-truth / detection methods. But what is the feature extractor used on top of those image regions? Comparing resnet / densenet extracted features with vgg / googlenet feature is not fair. - The presentation of this paper can be further improved. E.g. paragraph 2 in intro section is a bit verbose. Also breaking down overly-long sentences into shorter but concise ones will improve fluency. Some additional comments: - Figure 3: class semantic feature should be labeled as "s" instead of "c"? - equation 1: how v_G is fused from V_I? please specify. - equation 5: s is coming from textual representations (attribute / word to vec / PCA'ed TFIDF). It might have positive / negative values? However the first term h(W_{G,S}, v_G) is post ReLU and can only be non-negative? - line 157: the refined region vector is basically u_i = (1 + attention_weight) * v_i. since attention weight is in [0, 1] and sums up to 1 for all image regions. this refined vector would only scales most important regions by a factor of two before global pooling? Would having a scaling variable before attention weight help? - line 170: class semantic information is [not directly] embedded into the network? - Equation 11: v_s and u_G are both outputs from trained-network, and they are not normalized? So minimize L-2 loss could be simply reducing the magnitude of both vectors? - Line 201: the dimensionality of each region is 512: using which feature extractor? - Section 4.2.2: comparing number of attention layers is a good experiment. Another baseline could be not using Loss_G? So attention is only guided by global feature vector. - Table 4: what are the visual / textual representations used in each method? otherwise it is unclear whether the end-to-end performance gain is due to proposed attention model.